# *Cannabis sativa* L. Extract Alleviates Neuropathic Pain and Modulates CB1 and CB2 Receptor Expression in Rat

**DOI:** 10.3390/biom14091065

**Published:** 2024-08-26

**Authors:** Joanna Bartkowiak-Wieczorek, Agnieszka Bienert, Kamila Czora-Poczwardowska, Radosław Kujawski, Michał Szulc, Przemysław Mikołajczak, Anna-Maria Wizner, Małgorzata Jamka, Marcin Hołysz, Karolina Wielgus, Ryszard Słomski, Edyta Mądry

**Affiliations:** 1Physiology Department, Poznan University of Medical Sciences, 60-781 Poznan, Poland; emadry@ump.edu.pl; 2Department of Clinical Pharmacy and Biopharmacy, Poznan University of Medical Sciences, 60-781 Poznan, Poland; agbienert@ump.edu.pl (A.B.); anna.usciniak@gmail.com (A.-M.W.); 3Department of Pharmacology, Poznan University of Medical Sciences, 60-806 Poznan, Poland; kczora@ump.edu.pl (K.C.-P.); radkuj@ump.edu.pl (R.K.); mszulc@ump.edu.pl (M.S.); przemmik@ump.edu.pl (P.M.); 4Department of Pediatric Gastroenterology and Metabolic Diseases, Poznan University of Medical Sciences, 60-572 Poznan, Poland; mjamka@ump.edu.pl (M.J.); kwielgus@ump.edu.pl (K.W.); 5Department of Biochemistry and Molecular Biology, Poznan University of Medical Sciences, 60-781 Poznan, Poland; mholysz@ump.edu.pl; 6Department of Biotechnology, Institute of Natural Fibres and Medicinal Plants National Research Institute, 60-630 Poznan, Poland; slomski@up.poznan.pl

**Keywords:** *Cannabis sativa*, cannabidiol (CBD), Δ9-tetrahydrocannabinol (THC), vincristine, gabapentin, gene expression

## Abstract

Introduction: *Cannabis sativa* L. (CSL) extract has pain-relieving potential due to its cannabinoid content, so the effects of two CSL extracts on alleviating neuropathic pain were investigated in vivo. Methods and groups: Male Wistar rats (n = 130) were divided into groups and received vincristine (0.1 mg/kg) and gabapentin (60 mg/kg) to induce and relieve neuropathic pain or CSL extracts (D and B). The mRNA and protein expression of the cannabinoid receptors type 1 and 2 (CB1R, CB2R) were evaluated in the cerebral cortex, hippocampus, and lymphocytes. Behavioural tests (Tail-Flick and von Frey) were performed on all animals. Results: VK-induced neuropathic pain was accompanied by decreased CB1R protein level and CB2R mRNA expression in the cortex. Gabapentin relieved pain and increased CB1R protein levels in the hippocampus compared to the vincristine group. Hippocampus CB1R protein expression increased with the administration of extract D (10 mg/kg, 40 mg/kg) and extract B (7.5 mg/kg, 10 mg/kg) compared to VK group. In the cerebral cortex CSL decreased CB1R protein expression (10 mg/kg, 20 mg/kg, 40 mg/kg of extract B) and mRNA level (5 mg/kg, 7.5 mg/kg of extract B; 20 mg/kg of extract D) compared to the VK-group.CB2R protein expression increased in the hippocampus after treatment with extract B (7.5 mg/kg) compared to the VK-group. In the cerebral cortex extract B (10 mg/kg, 20 mg/kg) increased CB2R protein expression compared to VK-group. Conclusion: Alterations in cannabinoid receptor expression do not fully account for the observed behavioural changes in rats. Therefore, additional signalling pathways may contribute to the initiation and transmission of neuropathic pain. The *Cannabis* extracts tested demonstrated antinociceptive effects comparable to gabapentin, highlighting the antinociceptive properties of *Cannabis* extracts for human use.

## 1. Introduction

Neuropathic pain is caused by damage or dysfunction of the somatosensory part of the nervous system [1,2] and is characterised by the formation of pathological foci of stimulation in the peripheral and central nervous systems. Its characteristic symptoms are associated with inflammation or nerve damage during disease (e.g., diabetes, HIV infection, Herpes Zoster, cancer, multiple sclerosis, Parkinson’s disease) or treatment (surgical intervention, chemo-radiotherapy) [3,4,5] but to date, no effective treatment regimens for neuropathic pain have been developed [6]. Gabapentin has shown the greatest efficacy in alleviating this pain [7].

*Cannabis sativa* L. (CSL) has pleiotropic therapeutic effects, alleviating neuropathic and inflammatory pain, stimulating appetite, antiemetic and antispasmodic effects, and alleviating symptoms of epilepsy, glaucoma, asthma, and many other diseases or accompanying symptoms. *Cannabis sativa* contains 483 compounds, of which 66 are classified as cannabinoids. The most well-known is Δ9-THC, which has psychoactive properties, analgesic (antinociceptive), anti-inflammatory, euphoric, antioxidant, and antiemetic effects. Cannabidiol (CBD) and CBD derivatives are known for their anxiolytic, antipsychotic, antinociceptive, antispasmodic, anti-inflammatory, and antioxidant properties. Cannabidiolic acid (CBDA) exhibits antibiotic properties [8], with cannabigerol (CBG), cannabigerolic acid (CBGA), and cannabichromene (CBC) exhibiting antibacterial, antifungal, antinociceptive, and anti-inflammatory effects. Cannabis also contains other compounds [9], and the opposing activities of these compounds result in a range of beneficial pharmacological possibilities arising from their mutual relationships, synergy, and dose-dependent biological activities [10].

The adverse effects and high risk of addiction associated with the use of conventional antinociceptive agents, such as nonsteroidal anti-inflammatory drugs (NSAIDs) and opioids, have driven researchers to seek alternative plant-based substances that can modulate the antinociceptive effects of synthetic drugs [11]. One promising candidate with well-documented antinociceptive properties in the literature is cannabis and its constituent cannabinoids [12], including cannabidiol (CBD), whose efficacy in pain relief has been confirmed in numerous scientific studies [13].

The active compounds found in cannabis, particularly cannabinoids, exert antinociceptive effects by interacting with the endocannabinoid system, specifically the CB1(CB1R) and CB2 (CB2R) receptors [14].

The antinociceptive effects of cannabinoids are associated with the cannabinoid receptors CB1R and CB2R. CB1R is the most abundant postsynaptic receptor in the central nervous system responsible for memory, motor coordination, and pain transmission [15]. It also influences the release of GABA and glutamate in the gastrointestinal tract, autonomic nerves, and blood vessels [16] and is present in many immune cells [17]. CB2R was initially located and highly expressed in the spleen and thymus, but subsequent pharmacological and molecular studies have shown that these receptors are also present in peripheral and central neurons. CB2R receptors are located post-synaptically in the cell bodies and dendrites of neurons in both the central and peripheral nervous systems [18], and their activation does not induce significant psychoactive effects, meaning it does not lead to narcotic effects. The presence of CB2R in the brain and spinal cord suggests their role in regulating cerebral blood flow and blood-brain barrier permeability [19]. Δ9-THC is the most important psychoactive agonist for CB1R [20], whereas cannabidiol has a low affinity for CB1R/CB2R receptors. CBD non-competitively antagonises CB1R agonists at nanomolar concentrations [21] and is also an inverse agonist of the CB2R receptor. It has been demonstrated that CB2R responds to pain triggered by various stimuli, including thermal and mechanical activation, systemic inflammation, and tactile and thermal hypersensitivity characteristic of neuropathic pain [22,23]. Although numerous clinical studies on the effects of cannabinoids have already been conducted [24], the pleiotropic nature and complexity of the biochemical and molecular mechanisms underlying the observed pharmacological effects, particularly in relation to pain, including neuropathic pain, still require further preclinical analysis in animal models.

Available studies suggest a significant role of the TRPV1 receptor in regulating pain transmission, including neuropathic pain. TRPV1, a key nociceptor, can be activated by cannabinoids such as CBD, leading to antinociceptive effects similar to capsaicin. This mechanism involves opening the TRPV1 ion channel, allowing the influx of calcium and sodium ions to transmit the pain signal to the brain [25]. However, it is currently unclear whether the TRPV1 receptor alone is responsible for pain signal transmission or if other receptors, such as CB1R and CB2R, are also involved. The scientific literature in this area remains inconclusive, highlighting the need for further research to fully understand the mechanisms that regulate pain perception [26]. Cannabinoids regulate pain processing independently of opiates [27] in various animal models of neuropathic pain [28,29,30,31,32]. Therefore, it is crucial to determine whether CB1R and CB2R are involved in nociception.

Our study aims to clarify the expression of these receptors at the mRNA and protein levels, contributing to a better understanding of their role in pain transmission mechanisms. To this end, we investigated the pain-relieving effects of extracts derived from two varieties of *Cannabis sativa* L. (CSL) in a rat model of neuropathic pain.

## 2. Methods

### 2.1. Cannabis sativa L. (CSL) Extract Preparation

Hemp was cultivated in experimental plots (Institute of Natural Fibres and Medicinal Plants, Poznan, Poland) at a seed sowing density of 30 kg of seeds/ha and fertilised with nitrogen fertiliser (30 kg/ha). The panicles were harvested at the late flowering stage when they contain the highest cannabinoid content, as described by Szalata et al. [33]. The two-phase solvent extraction exposed the plant material to an organic solvent at 30 °C and then concentrated the resulting extract in an evaporator (50 mbar vacuum). Subsequently, the extract was dissolved in a mixture of ethanol and water at 80 °C, evaporated, and concentrated in an evaporator (50 mbar vacuum) before decarboxylation at 130 °C.

### 2.2. Characteristics of the CSL Extracts

The extracts were characterised by HPLC, as described by Szalata et al. [33]. The composition of KC Dora, a Hungarian variety of fibrous monoecious hemp bred at Agromag Kft, designated as Hemp flower extract version D, is shown below.


**Component**

**CBD-A**

**CBD**

**Δ9-THC-A**

**Δ9-THC**
Concentration [mg/g]1.2215.20.1513.3

The hemp variety Tygra, designated as Hemp flower extract version B, derived from a Polish strain of industrial hemp monoecious cultivated at the Institute of Natural Fibers and Medicinal Plants, has the following composition:


**Component**

**CBD-A**

**CBD**

**Δ9-THC-A**

**Δ9-THC**
Concentration [mg/g]1.2220.20.115.5

### 2.3. Cannabinoid Identification and Quantification

Extract samples (150 mg) were dissolved in 10 mL of a methanol (MeOH) and tetrahydrofuran (THF) mixture, shaken for 30 min, and then centrifuged at 5500 rpm. The supernatant was diluted tenfold and transferred to an HPLC vial for HPLC analysis using a gradient of 0.1% phosphoric acid (H_3_PO_4_) in water with acetonitrile (CAN) on a C18 stationary phase column with UV light absorbance monitored at 230 nm. Cannabinoid identification and quantification were conducted based on standard chromatograms as described by Zielińska et al. [34].

### 2.4. Animal Study

Pharmacological studies were conducted in collaboration with the Department of Pharmacology at the University of Medical Sciences in Poznan. The study protocol was approved by the local Ethics Committee for Animal Research (consent number: Resolution No. 42/2015 of the local Ethics Committee for Animal Research in Poznań) and conducted according to current guidelines for the care of laboratory animals and ethical guidelines for investigations of experimental pain in conscious animals.

Five male Wistar rats (200–250 g) were housed per cage at a constant temperature (22 ± 2 °C) with a 12:12 h light/dark cycle and free access to food and water at all times for at least a week before the experiment. Neuropathic pain was induced by the intraperitoneal (i.p.) administration of vincristine (0.1 mg/kg body weight) for five days, followed by two days of an equivalent volume of saline solution. This cycle was repeated twice before gabapentin (60 mg/kg bw) alone and with the CSL extracts and CBD administered to the stomach for five days. The assessment of the basal nociception threshold before administering the tested substances (vincristine, gabapentin, *Cannabis sativa* extracts) was not performed. As a result, we did not evaluate the presence or impact of any pre-existing nociception in the vincristine-treated groups. Instead, comparisons were made solely between the groups treated with these substances and the NaCl control group. This control group (NaCl group), which served as a negative control (where animals received NaCl instead of substances inducing (vincristine) or influencing (gabapentin) a pathological condition), did not experience neuropathic pain. Animals in the control group were also administered the test substances, allowing for a precise evaluation of their effects without induced pain. The doses were calculated in relation to their cannabidiol content (5, 7.5, 10, 20, and 40 mg/kg) or an appropriate volume of vehicle, and the treatment groups are described in Table 1.

After the administration of the CSL extracts, the rats were decapitated to harvest the brain tissue and collect the blood.

#### 2.4.1. Induction of Neuropathic Pain

To induce neuropathic pain, animals, except for the control group, were administered VC i.p. at a dose of 0.1 mg/kg for five days, followed by a corresponding volume of saline i.p. for two days; this cycle was repeated twice [35]. Animals were then treated with studied substances for 5 days, and a tail-flick test was performed. The rats were then treated for two days, the last time the following day, after which the von Frey test was performed. The experimental scheme is presented in Figure 1.

#### 2.4.2. Tail-Flick Test

This test was designed to evaluate the antinociceptive effect by measuring the time of onset of the pain response to a thermal stimulus. A beam of focused light was directed at the rat’s tail, and the time from the placement of the rat on the apparatus in the appropriate position until the animal voluntarily takes (lifts) its tail from the field of incidence of the light beam was measured. Five consecutive measurements were recorded for each animal to calculate the mean immediately before the administration of the substance (time t = 0). Subsequent measurements were recorded 1, 2, 3, and 6 h after administration. The maximum measurement time was set at 60 s due to possible tissue damage if the rat did not respond [36].

#### 2.4.3. The von Frey Test

This test assessed sensory sensitivity by measuring the time of onset of a pain response to a mechanical stimulus. The animal was placed in a transparent plastic cage with a 0.5 cm wire mesh backing and then placed in the device for 3 min before the measurement to adapt to the conditions. Measurements (3–6) of a gradual increase in the pressure force of a metal filament with a diameter of 0.5 mm on the sole part of the rat’s hind right paw were recorded. The increase in the force of the filament ranged from 0 to 50 g for 10 s, and the endpoint of the measurement was the paw retraction. Up to 6 measurements were recorded immediately before the administration of the substance (time t = 0) and 1, 2, 3 and 6 h after administration to calculate the mean [36].

### 2.5. Real-Time PCR

Lymphocytes were isolated from peripheral blood using gradisol. Total RNA was isolated from the rat brain tissue homogenates (frontal cortex, hippocampus) and blood lymphocytes using TriPure Isolation Reagent (Roche Applied Science, Berlin, Germany) according to the manufacturer’s protocol [37] and reverse transcribed using the Transcriptor First Strand cDNA Synthesis Kit (Roche Applied Science, Berlin, Germany). Changes in CB1R and CB2R mRNA expression in the hippocampus, frontal cortex, and peripheral blood lymphocytes were quantified by real-time PCR using a LightCycler TM (Roche Applied Science, Berlin, Germany), a LightCycler Fast Start DNA Master SYBR Green I kit (Roche Applied Science, Berlin, Germany) and gene-specific primers (Table 2).

### 2.6. CNRI and CNR2 ELISA

CNR1 and CNR2 were quantified in rat tissue homogenates, cell lysates, and other biological fluids by ELISA. Tissue homogenates were prepared by rinsing the tissue samples in ice-cold PBS (0.01 mol/L, pH 7.0–7.2) to remove excess blood and homogenisation in 5–10 mL of PBS using a glass homogeniser on ice. The homogenates were centrifuged for 5 min at 5000× *g*, and the total protein content was determined using the BCA (bicinchoninic acid) method. The tissue homogenate samples and CNR1/CNR2 standards were added to a 96-well plate and allowed to bind to the immobilised antibodies before the absorbance was measured to generate a standard curve to determine the CNR1 and CNR2 concentrations.

### 2.7. Statistical Analysis

The data were statistically analysed using the Statistica 13.0 software (TIBCO Software Inc., Palo Alto, CA, USA), and a p-value of less than 0.05 indicated statistical significance. The Shapiro-Wilk test was applied to assess the normality of the data, and owing to the non-normal distribution of many variables, the descriptive statistics are presented as the median and interquartile range (IQR). The Mann-Whitney U test was used to compare two groups, while the Kruskal-Wallis test, followed by Dunn’s post-hoc test, was employed for comparisons involving more than two groups. The results for the Tail Flick and Von Frey tests are presented as mean ± SEM and analysed using the Statistica 12 program. Repeated measures analysis of variance (ANOVA) was applied for dependent and independent variables. Following ANOVA, Fisher’s Least Significant Difference (LSD) test was conducted as a post-hoc analysis to determine the significance of differences between specific groups.

## 3. Results

### 3.1. Neuropathic Pain Induction and Treatment

Two behavioural tests, the tail-flick test, and the von Frey test, were performed to confirm the induction of neuropathic pain in rats (Figure 2, Table 3). There were differences between the vincristine (VK) and NaCl groups at all time points in the tail-flick test, with a significant increase in pain sensitivity observed in the vincristine (VK) group compared to the control animals at all time points examined in the von Frey test.

The antinociceptive effect of gabapentin (GP) was confirmed in the tail-flick and von Frey tests in rats receiving vincristine and gabapentin (VK + GP) compared to the group receiving vincristine (VK; Figure 3, Table 4). There was a significant difference between the WK and WK + GP groups after 6 h of gabapentin administration; however, the response to the applied stimulus began to increase after 3 h of drug administration in the tail flick test. In the group of animals receiving VK and GP, at the sixth hour after the administration, the effect was also statistically significant at the initial measurement point (T0) relative to the VK group (*p* < 0.05). There was a significant difference between the WK + GP and WK groups at 3 and 6 h after gabapentin administration in the von Frey test.

A comparison of the antinociceptive effect in the tail-flick test in rats receiving vincristine and extract D (VK + CBD D) or extract B (VK + CBD B) versus the group receiving vincristine (VK) is shown in Figure 4 and Table 5. In the tail-flick test, animals receiving extract B showed an increase in the response time to thermal stimulus, with the most significant results observed after 1 h of administration for the extract given at doses of 5 mg/kg and 20 mg/kg. A statistically significant effect was noted only after 6 h following the highest dose 40 mg CBD/kg. Compared to VK the statistically significant increased reaction time to a pain stimulus (an antinociceptive effect) was observed especially in the 20 mg/kg dose. For extract D, statistically significant differences were observed between the VK + CBD (D) and control (VK) groups at the lowest dose of 5 mg/kg (after 1, 2, 3, 6 h), at the dose of 7.5 mg/kg (after 1 and 2 h post-administration), and at 10 mg/kg (after 1-h post-extract administration). This effect (VK + CBD (D) vs. VK) was significant not only at the given time points but also with respect to the initial measurement point (T0). Moreover, at the dose 20 mg/kg and the highest dose (40 mg/kg), the extract exhibited significant antinociceptive effects after 6 h compared to the baseline (t = 0).

A comparison of the antinociceptive effects in the von Frey test was also conducted in rats receiving vincristine and extract D (VK + CBD D) and vincristine and extract B (VK + CBD B) relative to the group receiving vincristine (VK) (Figure 5, Table 6). In the von Frey test, statistically significant differences compared to the group receiving vincristine were observed for extract D at doses of 7.5 mg/kg, 10 mg/kg, 20 mg/kg, and 40 mg/kg; this effect also occurred at T0 (the animals had already received the extract for 7 days). The most significant increase in response was noted 1 h after administration of doses of 10 mg/kg, 20 mg/kg, and 40 mg/kg. A significant increase in antinociceptive effect was also observed 2 h after administration for doses of 7.5 mg/kg and 10 mg/kg, and 3 h after administration for the dose of 7.5 mg/kg.

For extract B, statistically significant differences compared to vincristine in the von Frey test were observed at a dose of 20 mg/kg after 1, 2, and 3 h (in the case of the last two measurement points—only vs. T0). Additionally, significant antinociceptive effects were observed 1 h after administration of the lowest dose (5 mg/kg) and 2 h after administration of 10 mg/kg.

### 3.2. The Influence of Neuropathic Pain Induction and Treatment on CB1R and CB2R Protein and mRNA Expression

#### 3.2.1. Vincristine and Gabapentin Treatment on CB1R and CB2R Expression

There was a significant decrease in CB1R protein expression and CB2R mRNA expression in the cerebral cortex compared to the group receiving NaCl after vincristine-induced neuropathic pain, whereas there was a significant increase in CB1R protein levels in the hippocampus compared to the group receiving vincristine after the administration of gabapentin (Table 7).

#### 3.2.2. CB1R Expression in the Rat Brain and Blood Lymphocytes

Significant alterations in CB1R expression were observed in the brain and blood lymphocytes on neuropathic pain induction (Table 8). Hippocampus CB1R protein expression increased with the administration of extract D at a dose of 10 mg/kg and 40 mg/kg. Additionally, the groups receiving extract D in dose 10 mg/kg (VK + Extract D 10 mg/kg) and 40 mg/kg (VK + Extract D 40 mg) differed significantly from those receiving this extract in dose 5 mg/kg (VK + Extract D 5 mg/kg).

Similarly, extract B, particularly at doses of 7.5 mg/kg and 10 mg/kg, led to significant increases in CB1R protein expression in the hippocampus compared to the vincristine group—VK + Rape oil. There were no statistically significant changes in CB1R mRNA expression in the hippocampus compared to the group receiving vincristine.

However, the 7.5 mg/kg dose of extract D (VK + Extract D 7.5 mg/kg) caused a statistically significant variation in CB1R expression at the mRNA level in the hippocampus compared to the 10 mg/kg dose of this extract (VK + Extract D 10 mg/kg).

Only extract B at doses of 10 mg/kg, 20 mg/kg, and 40 mg/kg decreased CB1R protein expression in the cerebral cortex compared to the vincristine group—VK + Rape oil. Extract D at a dose of 20 mg/kg and extract B at doses of 5 mg/kg and 7.5 mg/kg reduced CB1R mRNA expression in the cerebral cortex compared to the group receiving vincristine.

However, the level of CB1R protein expression in the cerebral cortex differed significantly between the groups receiving extract D at a dose of 7.5 mg/kg compared to animals receiving extract D at a dose of 40 mg/kg, as well as between the group receiving 10 mg/kg and those receiving 20 mg/kg and 40 mg/kg of this extract. At the mRNA level, the 7.5 mg/kg and 20 mg/kg doses of Extract D also differed significantly from each other (VK + Extract D 7.5 mg/kg vs. VK + Extract D 20 mg/kg).

No statistically significant changes in CB1R expression in lymphocytes were observed compared to the group receiving vincristine. Statistically significant differences at the mRNA level were observed only between groups receiving extract D at 5 mg/kg and 10 mg/kg doses (VK + Extract D 5 mg/kg vs. VK + Extract D 10 mg/kg).

### 3.3. CB2R Expression in the Rat Brain and Blood Lymphocytes

CB2R protein and mRNA expression changed in the hippocampus and cerebral cortex (Table 9). CB2R protein expression increased in the hippocampus after treatment with extract B at a dose of 7.5 mg/kg compared to the group receiving vincristine (VK).

At the mRNA level, the dose of 7.5 mg/kg of extract D (VK + Extract D 7.5 mg/kg) caused a statistically significant variation in CB1R expression at the mRNA level in the hippocampus compared to the 10 mg/kg dose of this extract (VK + Extract D 10 mg/kg).

In the cerebral cortex, doses of 10 mg/kg and 20 mg/kg of extract B increased CB2R protein expression compared to the group receiving vincristine (VK). Additionally, at the protein level, the group receiving extract B at a dose of 10 mg/kg (VK + Extract B 10 mg/kg) differed significantly from the group receiving the extract at a dose of 40 mg/kg (VK+ Extract B 40 mg/kg). At the mRNA level, the 10 mg/kg and 20 mg/kg doses of Extract D differed significantly from each other (VK + Extract D 10 mg/kg vs. VK + Extract D 20 mg/kg) (Table 4).

## 4. Discussion

The structures responsible for processing and perceiving pain in the brain are components of the limbic system, including the primary and secondary somatosensory cortices (S1/S2), prefrontal cortex (PFC), anterior cingulate cortex (ACC), amygdala (AMG), and nucleus accumbens (NAc). CB1R is found in structures of the central and peripheral nervous systems implicated in pain processing, such as the amygdala, brainstem, spinal cord, and peripheral sensory neurons [38]. For the molecular analysis of neuropathic pain in this study, samples were taken from the cerebral cortex and the hippocampus, both of which are integral parts of the limbic system. However, the precise mechanisms behind the heightened sensitivity of the endogenous cannabinoid system in chronic pain, particularly in the brain, remain incompletely understood. Furthermore, the complex nature of hyperalgesia involves multiple factors [28]. Among the known physiological changes and signalling pathways contributing to neuropathic pain development, alterations in CB1R and CB2R expression have been noted [39,40], with both CB1R and CB2R implicated in inflammatory and neuropathic pain development [41,42,43]. The neurobiological mechanisms driving the heightened sensitivity of the endogenous cannabinoid system in chronic pain remain poorly understood.

We conducted our study on male Wistar rats based on widely accepted literature indicating that males are the most commonly used gender in neuropathic pain research [31,42,43,44,45]. The choice of gender was also driven by the fact that male subjects are more hormonally predictable compared to females, which eliminates the need to adjust the experimental procedure according to different phases of the reproductive cycle. Using males also minimizes potential behavioural changes related to hormonal fluctuations, thereby increasing the consistency and repeatability of the results.

One of the adverse effects of chemotherapeutic agents, such as vincristine, is peripheral neuropathic pain [46]. Based on in vivo studies conducted using a rat model, a dose of 0.1 mg/kg vincristine has effectively induced peripheral neuropathy [47,48,49,50]. In the present study, neuropathic pain was induced using vincristine (VK) at a dose of 0.1 mg/kg (i.p.) [51,52].

Gabapentin is a recommended treatment for neuropathic pain, with its efficacy having been repeatedly confirmed in in vivo studies using male Wistar rats. In our study, gabapentin was administered at a dose of 60 mg/kg body weight, which is consistent with the literature that identifies this dose as effective in models of chemotherapy-induced neuropathic pain [44,45].

The changes in response to thermal and mechanical stimuli in animals administered vincristine compared to the control confirmed the effective induction of neuropathic pain in rats. The neuropathic pain analyses indicated that the expression of CB1R increases in areas where pain transmission signal pathways are present [53]. Increased CB1R mRNA and protein expression in the peripheral nerve fibres of the hind paw and spinal ganglia has been reported in an experimental model of inflammatory and neuropathic pain induced by complete Freund’s adjuvant (CFA) and neuronal damage in Male Sprague Dawley rats [53]. The use of a control group in our study, specifically a negative control (where animals received NaCl instead of substances inducing a pathological condition), aimed to reveal behavioural responses and molecular changes in animals that were not subjected to nociception-induced by pharmacological agents and, thus, by design, did not experience neuropathic pain. Animals in the control group were also administered the test substances, allowing for a precise evaluation of their effects without induced pain. We observed that administration of vincristine to induce neuropathic pain was associated with a decrease in the expression of CB1R and CB2R in the cerebral cortex at both the protein and mRNA levels compared to the control group receiving NaCl. However, there were no statistically significant changes in the CB1R and CB2R in the hippocampus and lymphocytes.

CB1R levels increased in the superficial dorsal horn of the spinal cord of rats with neuropathic pain following sciatic nerve injury [54] in contrast to the findings reported by Lim et al. However, it is noteworthy that the observed changes in CB1R expression were localised to the site of damage [54], and we evaluated CB1R levels in the brain following systemic neuropathy induction.

The changes in response to stimuli in animals administered vincristine and gabapentin (60 mg/kg, p.o.) compared to the group that received only vincristine indicate that the pain induced by vincristine is sensitive to treatments used for alleviating neuropathic pain [39]. An explanation is needed, especially why gabapentin administered chronically in our experiments showed efficacy only in the final hours after administration (i.e., from the third hour). However, this is consistent with published observations that noted that this drug, at the same dose used in our study, begins to take effect after at least 5 days of administration and only 4 h after the given dose [55]. Interestingly, gabapentin administration in our study caused a significant increase in CB1R protein levels in the hippocampus compared to the group receiving vincristine (VK).

Our study used two tests to assess hyperalgesia: the hot plate test and the tail flick test. It is important to note that there has been no conclusive evidence to suggest that cannabinoid receptors can be activated by thermal hyperalgesia. However, the perception of pain and temperature is closely linked through the action of specific receptors that respond to thermal stimuli. TRPV1 receptors, for instance, respond to high temperatures perceived as painful. These receptors are activated by thermal stimuli such as heat and are responsible for inducing pain sensations. TRPV1 is part of a larger family of TRP receptors, which play a crucial role in converting thermal information into nerve impulses that are subsequently interpreted by the brain as pain sensations [56].

There are differences in the effects of CB1R and CB2R antagonists depending on the time and level of administration and the type of neuropathic pain model, suggesting the complexity of pain regulation mechanisms in the endocannabinoid system [57].

The involvement of CB1R and CB2R in neuropathic pain modulation remains debatable. While some studies using selective receptor agonists and endocannabinoid system modulators confirmed receptor expression changes during neuropathic pain and post-agent use, evidence also suggests that cannabinoid antinociceptive effects may be independent of cannabinoid receptors and mediated by TRPV1 receptor [14]. Castan˜e et al. [58] reported that neuropathic pain behaviours developed similarly in mice lacking CB1Rs, confirming the efficacy of gabapentin in pain management even in the absence of CB1Rs.

Contrary to our observations, analyses of CB1R levels in the amygdala of rats with neuropathic pain revealed increased CB1R expression. Cannabinoid receptor CB1R expression in the amygdala was studied in a rat model of chronic neuropathic pain induced by unilateral transection of the sciatic nerve, showing an increase in CB1R expression in the amygdala on the first day after nerve damage, peaking within two days and returning to control levels after two weeks [59].

The group receiving vincristine served as a reference for comparing the antinociceptive effects of the extracts used in the present study: extract B and extract D of *Cannabis sativa*. At the behavioural level, both extracts reduced neuropathic pain perception in response to thermal and mechanical stimuli. The antinociceptive effect was more pronounced in the von Frey test compared to the thermal stimulus test, consistent with reported findings [51,60]. The two extracts differed in their antinociceptive efficacy, with extract D appearing to be more effective.

Costa et al. [61] showed that CBD at a dose of 20 mg/kg administered for 7 days reduced neuropathic pain. In our study, a dose of 20 mg/kg of extract B produced an antinociceptive effect after one hour in the tail-flick test, while the same dose of extract D showed antinociceptive effects in the von Frey test. Cannabinoids exhibit antinociceptive activity in rats’ chronic neuropathic and inflammatory pain models, primarily mediated by CB1 receptor activation [59,62]. The present study regarding CB1R and CB2R in the hippocampus and cerebral cortex following administration of various doses of *Cannabis sativa* extracts compared to the vincristine group (VK + Rape oil) indicates significant changes in cannabinoid receptor expression. Administration of both CSL extracts (extract D and B) increased CB1R and CB2R protein expression in the hippocampus, a region strongly implicated in neuropathic pain [63]. CB1R protein expression increased after extract D (20 mg/kg and 40 mg/kg) and extract B (7.5 mg/kg and 10 mg/kg) compared to vincristine group—VK + Rape oil. Existing research suggests that the hippocampus exhibits a high level of CB1R, especially in GABAergic neurons, which may affect interaction with cannabinoids. CB1R activation influences the pro-inflammatory response induced by stress at multiple levels. Increased CB1R expression in vitro and in vivo studies significantly reduced inflammation in neurons, astrocytes, and microglia [64,65,66]. Contrary to the results observed in the hippocampus, extract B (10 mg/kg, 20 mg/kg, 40 mg/kg) decreased CB1R protein expression in the cerebral cortex compared to the vincristine group—VK + Rape oil. Extract D (20 mg/kg) and extract B (5 mg/kg, 7.5 mg/kg) reduced CB1R mRNA expression in the cerebral cortex compared to the group receiving vincristine. CB2R protein expression increased in the hippocampus after treatment with extract B at a dose of 7.5 mg/kg compared to the group receiving vincristine. Similarly, doses of 10 mg/kg, and 20 mg/kg of extract B resulted in higher CB2R protein expression in the cerebral cortex compared to the group receiving vincristine.

The differential expression of CB1R and CB2R in the nervous system, cerebral cortex, and hippocampus may result from the distinct functions and locations of these receptors. CB1R, which are physiologically expressed centrally (dorsal horns and dorsal root ganglion) and peripherally [67], may have increased expression following nerve injury [68]. These receptors are present in neurons and brain regions responsible for movement, posture control, pain, sensory perception, memory, cognitive functions, emotions, autonomic functions, and the endocrine system. They operate by inhibiting neurotransmitter release, which may explain why antagonising CB1R can reverse gabapentin-induced loss of hypersensitivity [57]. However, CB2R in immune cells and keratinocytes may reduce the release of nociceptive factors or even increase the release of antinociceptive factors (opioids) [68]. Since receptor expression is a time-dependent process [69], this may explain the variability observed in our molecular and behavioural results. These results suggest that cannabinoid influence on brain processes may be complex and region-specific [70]. Understanding the connections between diverse cannabinoid receptor expression in the central nervous system and peripherally is crucial for comprehending CBD’s ambiguous effects [71]. Walker et al. [72] questioned the antinociceptive effects of cannabinoids. CBD analgesia independent of CB1R and CB2R activation, as observed by Castan˜e et al. [58], suggests the involvement of other signalling pathways and receptors in the modulation of inflammation and chronic neuropathic pain, such as TRPV1 [14], or the activation of glycine channels or CaV3.2 channels [62]. There is a lack of studies on the impact of *Cannabis sativa* extracts on the expression of CB1R and CB2R in the central and peripheral nervous systems, which may be due to difficulties in interpreting results because of the pleiotropic pharmacological activity of individual compounds, as well as interactions between compounds that may act synergistically or antagonistically within a single physiological effect. Additionally, not all active compounds in cannabis that may interact with cannabidiol (CBD) and tetrahydrocannabinol (THC) are currently known. Comelli et al. [73] highlighted the significant role of the synergy between cannabinoids and other compounds present in *Cannabis sativa* extract (e.g., terpenes or flavonoids). The complex interactions of *Cannabis* extracts and interactions between various compounds can lead to unexpected pharmacological effects compared to isolated active substances like CBD and THC. This phenomenon, known as pleiotropic interaction, influences the observed pharmacological effect of *Cannabis* extracts. Interestingly, our molecular expression analysis results at both the mRNA and protein levels do not confirm a dose-dependent effect of *Cannabis* extracts. The assumption that dose-dependency should result from pharmacological activity seems valid in the case of single active substances. However, for plant extracts, due to the presence of multiple active compounds, the observed pharmacological effect is not only variable but also biphasic and often independent of dose [74,75,76]. This variability extends to cannabinoid receptor expression levels, which we observed as independent of the dose used. Moreover, a primary function of the endocannabinoid system is to maintain homeostasis in the body, which involves appropriately modulating cannabinoid receptor activity as regulators of physiological processes [77]. Moreover, the pleiotropic mechanism of action of C. sativa extracts may also manifest itself in vitro. For CBD alone, it has been shown experimentally under neuronal culture conditions that it can induce the expression about 680 genes (THC for the variety only 58), and 524 gene products may be down-regulated (in the case of THC). The huge number of CBD-induced genes included mostly those controlled by nuclear factors known to be involved in the regulation of stress response and inflammation, mainly via the (EpRE/ARE)-Nrf2/ATF4 system and the Nrf2/Hmox1 axis [78]. How the individual studied components of the extracts interact dose-dependently with components of the complex orchestration of the intergenic network of interactions is a question that requires further, in-depth molecular investigations. The biphasic molecular effect of CBD in vitro in neurons was demonstrated in a study by Blando et al. in which the CB1 signalling pathway was responsible for neuronal differentiation at 10 µM via phosphorylated p-ERK and p-AKT kinases but not at 5 µM. At a lower dose, a new correlation between CBD, neurodifferentiation, and retinoic acid receptor-related orphan receptors (RORs) was observed [79]. Notably, cannabinoid receptors are metabotropic receptors, which engage secondary signalling pathways, various factors, and signal transducers. These components are simultaneously involved in numerous other cellular processes [80]. Given these considerations, the intricate nature of cannabinoid receptors and the diverse, pleiotropic effects of *Cannabis* extracts underscore the need for further comprehensive research. Despite these challenges, this complexity offers a promising avenue for discovering novel and exciting pharmacological properties of cannabinoids.

Study Limitations: Interpreting results regarding plant extracts always requires caution because, besides the basic active substances (e.g., CBD, THC), they contain a wide range of compounds that may simultaneously influence various signalling pathways and exert pleiotropic effects. However, the study of extracts is crucial because those using hemp oil or smoking dried leaves come into contact with the entire spectrum of substances contained in *Cannabis sativa* leaves. It should be noted that only the gene expression was assessed for lymphocytes due to limited material for protein isolation.

Summarising, vincristine-induced neuropathic pain was accompanied by downregulation of CB1R protein expression and CB2R mRNA expression in the cerebral cortex. Interestingly, gabapentin administration increased CB1R protein levels in the hippocampus compared to the vincristine group. Different doses of extract B increased CB1R protein levels in the hippocampus but decreased them in the cortex at higher doses (10 mg/kg, 20 mg/kg, and 40 mg/kg). Additionally, extract D (at 20 mg/kg) and lower doses of extract B (5 mg/kg and 7.5 mg/kg) reduced CB1R mRNA expression in the cortex. Notably, extract B enhanced CB2R protein expression in both the hippocampus (at 7.5 mg/kg) and cortex (at 10 mg/kg and 20 mg/kg).

## 5. Conclusions

Alterations in the cannabinoid receptors may not fully account for the behavioural changes observed in response to neuropathic pain, suggesting that additional signalling pathways could contribute to the initiation and transmission of neuropathic pain. The involvement of the TRPV1 receptor in analgesia and pain processing appears promising. Furthermore, both tested *Cannabis sativa* L. extracts demonstrated antinociceptive effects comparable to gabapentin, highlighting the potential medical value of *Cannabis* extracts for human use.

Our findings provide a foundation for further research in both in vitro and in vivo models to target specific cannabinoid receptors and the TRPV1 receptor to deepen our understanding of the receptor-based mechanisms underlying neuropathic pain influenced by *Cannabis*.

## Figures and Tables

**Figure 1 biomolecules-14-01065-f001:**
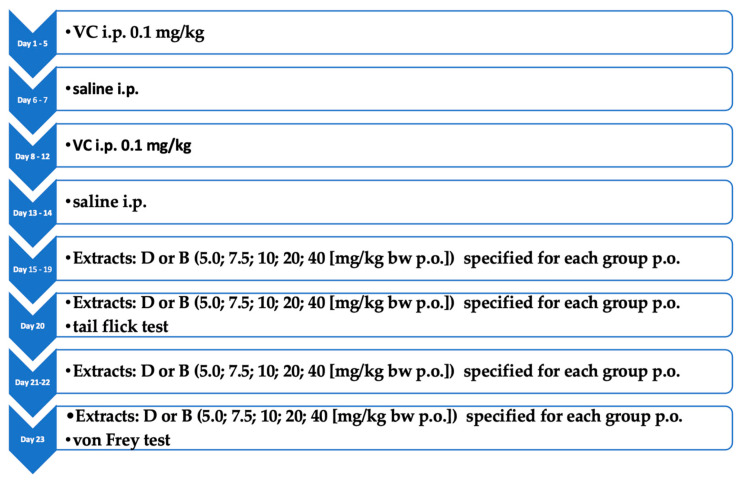
Experimental scheme of neuropathic pain induction and treatment in rats.

**Figure 2 biomolecules-14-01065-f002:**
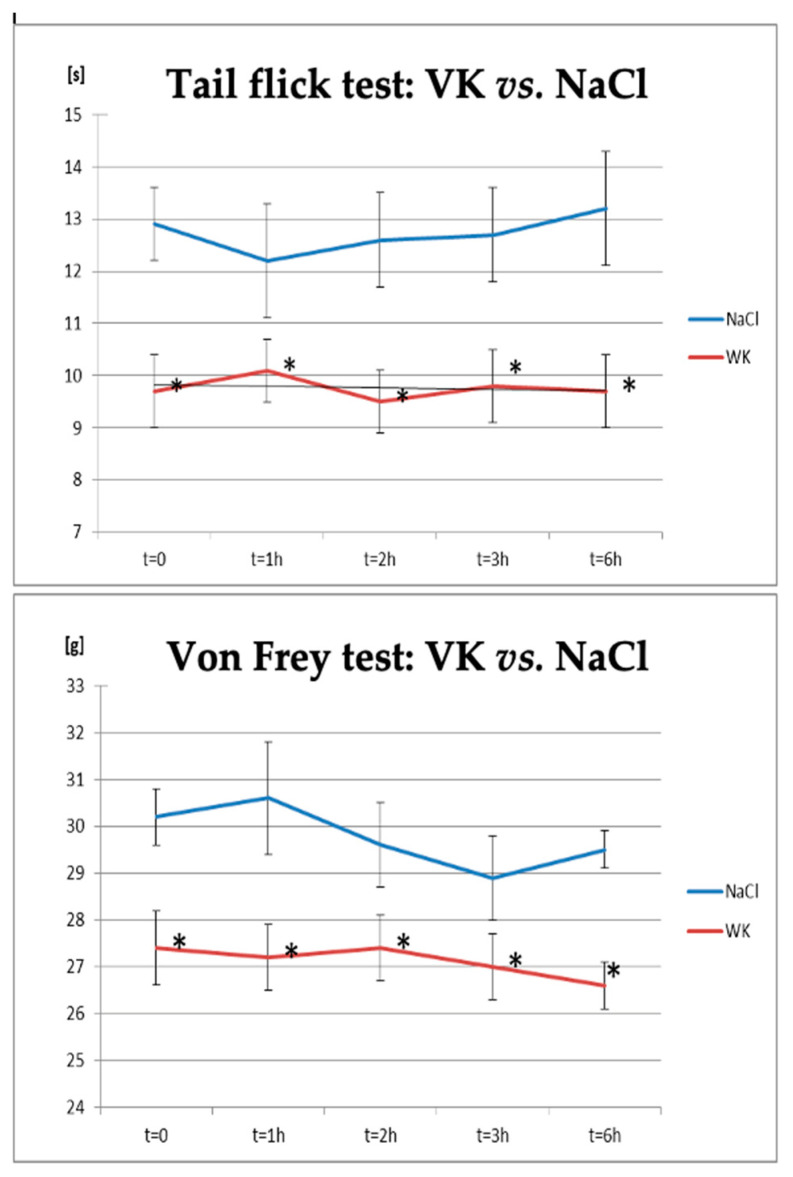
Changes in pain response in the tail-flick and von Frey tests in animals after administration of vincristine compared to the control group (NaCl). * statistically significant differences compared to the control group (NaCl; *p* < 0.05).

**Figure 3 biomolecules-14-01065-f003:**
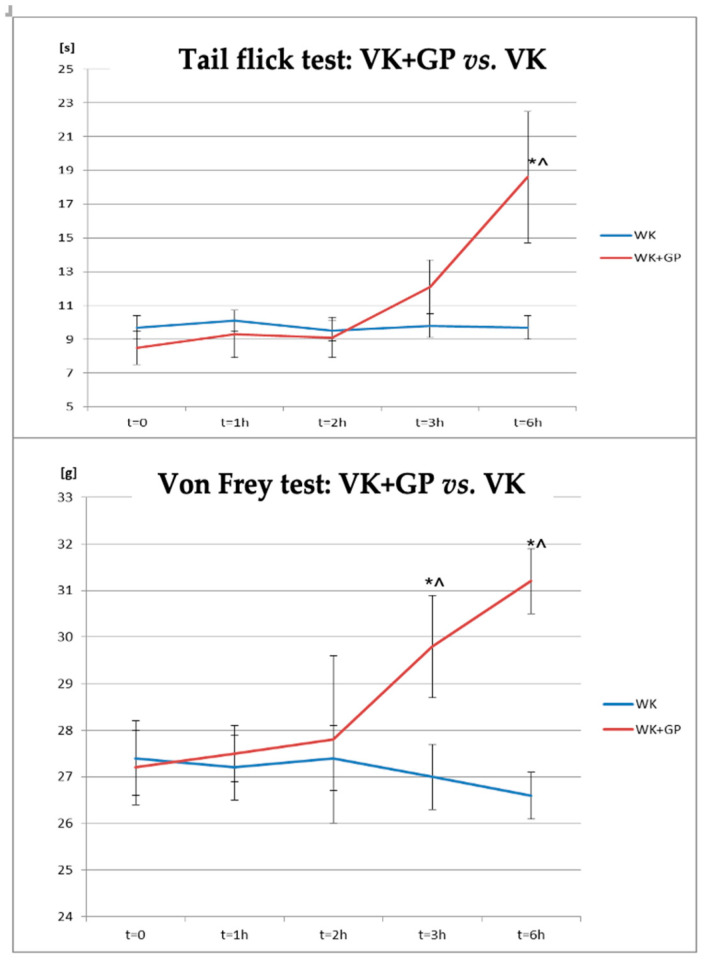
Changes in the pain response in the tail flick test and von Frey test in animals after combined administration of vincristine and gabapentin compared to the group of animals receiving vincristine alone. * statistically significant differences compared to VK (*p* < 0.05); ^—statistically significant differences for the respective group compared to t = 0 (*p* < 0.05).

**Figure 4 biomolecules-14-01065-f004:**
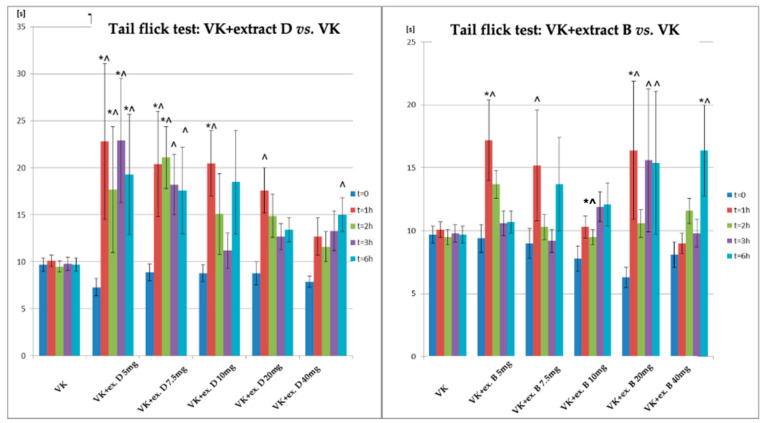
Changes in the pain response in the tail-flick test in animals after administration of vincristine and extract D, as well as vincristine and extract B, at different doses (5–40 mg/kg, p.o.) relative to the group receiving vincristine. * statistically significant differences compared to VK (*p* < 0.05); ^—statistically significant differences for the respective group compared to t = 0 (*p* < 0.05); VK—Vincristine—0.1 mg/kg of body weight, intraperitoneal administration; Extract D—*Cannabis sativa* L. extract, *variety Dora*; the dose expressed as synthetic Cannabidiol, oral administration Extract B—*Cannabis sativa* L. extract, *variety Tygra*; the dose expressed as synthetic Cannabidiol, oral administration i.p.—intraperitoneal; p.o.—per os.

**Figure 5 biomolecules-14-01065-f005:**
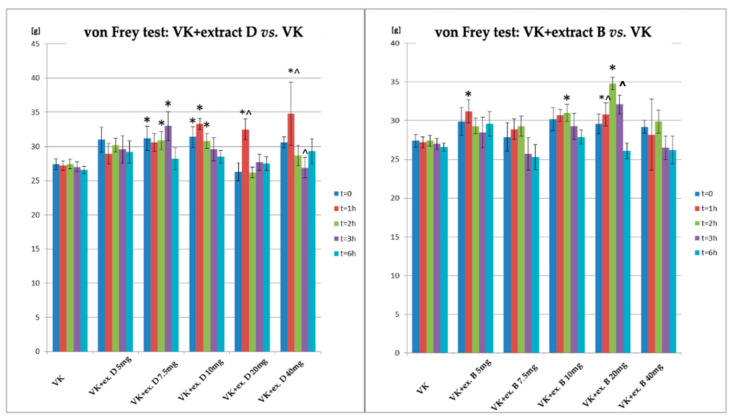
Changes in the pain response in the von Frey test in animals after administration of vincristine and extract D, as well as vincristine and extract B, at different doses (5–40 mg/kg, p.o.) relative to the group receiving vincristine. * statistically significant differences compared to VK (*p* < 0.05); ^—statistically significant differences for the respective group compared to t = 0 (*p* < 0.05); VK—Vincristine—0.1 mg/kg of body weight, intraperitoneal administration; Extract D—*Cannabis sativa* L. extract, *variety Dora*; the dose expressed as synthetic Cannabidiol, oral administration Extract B—*Cannabis sativa* L. extract, *variety Tygra*; the dose expressed as synthetic Cannabidiol, oral administration i.p.—intraperitoneal; p.o.—per os.

**Table 1 biomolecules-14-01065-t001:** The characteristics of the study groups and the interventions.

Group	Intervention	Number of Rats
1	NaCl (0.9%, 1 mL i.p.) + Rape oil (1 mL p.o.)	10
2	VK + Rape oil—1 mL p.o.	10
3	VK + Gabapentin (solution 1 mg/mL)—60 mg/kg bw p.o.	10
4	VK + Extract D—5.0 mg/kg bw p.o	10
5	VK + Extract D—7.5 mg/kg bw p.o	10
6	VK + Extract D—10.0 mg/kg bw p.o	10
7	VK + Extract D—20.0 mg/kg bw p.o	10
8	VK + Extract D—40.0 mg/kg bw p.o	10
9	VK + Extract B—5.0 mg/kg bw p.o	10
10	VK + Extract B—7.5 mg/kg bw p.o	10
11	VK + Extract B—10.0 mg/kg bw p.o	10
12	VK + Extract B—20.0 mg/kg bw p.o	10
13	VK + Extract B—40.0 mg/kg bw p.o	10

VK—Vincristine (Vincristine Sulfate, injection solution, 1 mg/mL)—0.1 mg/kg of body weight, intraperitoneal administration; Extract D—*Cannabis sativa* L. extract, *variety Dora*; the dose expressed as synthetic Cannabidiol, oral administration. Extract B—*Cannabis sativa* L. extract, *variety Tygra*; the dose expressed as synthetic Cannabidiol, oral administration. i.p.—intraperitoneal; p.o.—per os; bw—body weight.

**Table 2 biomolecules-14-01065-t002:** Primers used in Real-Time PCR.

Gene	Sequence 5′→ 3′
**CB1R**	Primer F: TAATATGAAGCAAGATACCAG Primer R: CCATTTACAGAGACAACAAG
**CB2R**	Primer F: CAGTTACAGAGACAGAGGC Primer R: TGTTTCCATTACCCTAGAGC
**GADPH**	Primer F: GATGGTGAAGGTCGGTGTG Primer R: ATGAAGGGGTCGTTGATGG

GAPDH was used as a housekeeping gene (endogenous internal standard) for normalisation.

**Table 3 biomolecules-14-01065-t003:** Changes in the pain response in the tail-flick test and von Frey test in animals after administration of vincristine compared to the control group (NaCl).

Group	t = 0	t = 1 h	t = 2 h	t = 3 h	t = 6 h
**Tail flick test**					
	[s]				
NaCl	12.9 ± 0.7	12.2 ± 1.1	12.6 ± 0.9	12.7 ± 0.9	13.2 ± 1.1
VK	9.7 ± 0.7 *	10.1 ± 0.6 *	9.5 ± 0.6 *	9.8 ± 0.7 *	9.7 ± 0.7 *
**Von Frey test**					
	[g]				
NaCl	30.2 ± 0.6	30.6 ± 1.2	29.6 ± 0.9	28.9 ± 0.9	29.5 ± 0.4
VK	27.4 ± 0.8 *	27.2 ± 0.7 *	27.4 ± 0.7 *	27 ± 0.7 *	26.6 ± 0.5 *

VK—Vincristine—0.1 mg/kg of body weight. * statistically significant differences compared to NaCl (*p* < 0.05).

**Table 4 biomolecules-14-01065-t004:** Changes in the pain response in the tail flick test and von Frey test in animals after combined administration of vincristine and gabapentin compared to the group of animals receiving vincristine alone.

Group	t = 0	t = 1 h	t = 2 h	t = 3 h	t = 6 h
**Tail flick test**					
	[s]				
VK	9.7 ± 0.7	10.1 ± 0.6	9.5 ± 0.6	9.8 ± 0.7	9.7 ± 0.7
VK + GP	8.5 ± 1	9.3 ± 1.4	9.1 ± 1.2	12.1 ± 1.6	18.6 ± 3.9 *^
**Von Frey test**					
	[g]				
VK	27.4 ± 0.8	27.2 ± 0.7	27.4 ± 0.7	27 ± 0.7	26.6 ± 0.5
VK + GP	27.2 ± 0.8	27.5 ± 0.6	27.8 ± 1.8	29.8 ± 1.1 *^	31.2 ± 0.7 *^

VK—Vincristine—0.1 mg/kg of body weight, intraperitoneal administration. GP—gabapentin. * statistically significant differences compared to NaCl (*p* < 0.05). ^ statistically significant differences compared to t = 0 (*p* < 0.05).

**Table 5 biomolecules-14-01065-t005:** Changes in the pain response in the tail-flick test in animals after administration of vincristine and extract D, as well as vincristine and extract B, at different doses (5–40 mg/kg, p.o.) relative to the group receiving vincristine.

Intervention	t = 0	t = 1 h	t = 2 h	t = 3 h	t = 6 h
			[s]		
VK + Gabapentin	9.7 ± 0.7	10.1 ± 0.6	9.5 ± 0.6	9.8 ± 0.7	9.7 ± 0.7
VK + Extract D 5 mg/kg	7.3 ± 0.9	22.8 ± 8.3 *^	17.7 ± 6.7 *^	22.9 ± 6.6 *^	19.3 ± 6.4 *^
VK + Extract D 7.5 mg/kg	8.9 ± 0.9	20.4 ± 5.6 *^	21.1 ± 3.3 *^	18.2 ± 3.2 ^	17.6 ± 4.6 ^
VK + Extract D 10 mg/kg	8.8 ± 0.9	20.5 ± 3.5 *^	15.1 ± 4.3	11.2 ± 1.9	18.5 ± 5.5
VK + Extract D 20 mg/kg	8.8 ± 1.2	17.6 ± 2.4^	14.9 ± 2.3	12.7 ± 1.4	13.4 ± 1.3
VK + Extract D 40 mg	7.9 ± 0.6	12.7 ± 2	11.6 ± 1.6	13.3 ± 2.1	15 ± 1.8 ^
VK + Extract B 5 mg/kg	9.4 ± 1.1	17.2 ± 3.2 *^	13.7 ± 1.1	10.6 ± 1.0	10.7 ± 0.9
VK + Extract B 7.5 mg/kg	9.0 ± 1.2	15.2 ± 4.4 ^	10.3 ± 1.0	9.2 ± 0.9	13.7 ± 3.7
VK + Extract B 10 mg/kg	7.8 ± 1.0	10.3 ± 0.9	9.5 ± 0.6 *^	11.9 ± 1.2	12.1 ± 1.7
VK + Extract B 20 mg/kg	6.3 ± 0.8	16.4 ± 5.5 *^	10.6 ± 1.1	15.6 ± 5.7 ^	15.4 ± 5.7 ^
VK + Extract B 40 mg/kg	8.1 ± 1.0	9.0 ± 0.8	11.6 ± 1.0	9.8 ± 1.1	16.4 ± 3.6 *^

* statistically significant differences compared to VK (*p* < 0.05); ^—statistically significant differences for the respective group compared to t = 0 (*p* < 0.05); VK—Vincristine—0.1 mg/kg of body weight, intraperitoneal administration Extract D—*Cannabis sativa* L. extract, *variety Dora*; the dose expressed as synthetic Cannabidiol, oral administration Extract B—*Cannabis sativa* L. extract, *variety Tygra*; the dose expressed as synthetic Cannabidiol, oral administration i.p.—intraperitoneal; p.o.—per os.

**Table 6 biomolecules-14-01065-t006:** Changes in the pain response in the von Frey test in animals after administration of vincristine and extract D, as well as vincristine and extract B, at different doses (5–40 mg/kg, p.o.) relative to the group receiving vincristine.

Grupa	t = 0	t = 1 h	t = 2 h	t = 3 h	t = 6 h
			[g]		
VK + Gabapentin	27.4 ± 0.8	27.2 ± 0.7	27.4 ± 0.7	27 ± 0.7	26.6 ± 0.5
VK + Extract D 5 mg/kg	31 ± 1.8	28.9 ± 1.5	30.2 ± 1.0	29.6 ± 2.0	29.2 ± 1.6
VK + Extract D 7.5 mg/kg	31.2 ± 1.8 *	30.6 ± 1.3	30.9 ± 1.3 *	33 ± 2.1 *	28.2 ± 1.6
VK + Extract D 10 mg/kg	31.4 ± 1.5 *	33.3 ± 0.8 *	30.8 ± 1.1 *	29.6 ± 1.7	28.5 ± 0.9
VK + Extract D 20 mg/kg	26.3 ± 1.3	32.5 ± 1.5 *^	26.2 ± 0.8	27.7 ± 1.2	27.5 ± 1.0
VK + Extract D 40 mg	30.6 ± 0.8	34.8 ± 4.6 *^	28.7 ± 1.5	26.9 ± 1.5 ^	29.3 ± 1.8
VK + Extract B 5 mg/kg	29.9 ± 1.3	31.2 ± 2.1 *	29.3 ± 1.9	28.5 ± 1.2	29.6 ± 2.3
VK + Extract B 7.5 mg/kg	27.9 ± 1.2	28.9 ± 1.2	29.3 ± 1.4	25.7 ± 2.0	25.3 ± 1.8
VK + Extract B 10 mg/kg	30.2 ± 1.4	30.7 ± 2.0	31 ± 1.8 *	29.3 ± 2.0	27.9 ± 1.5
VK + Extract B 20 mg/kg	29.6 ± 1.5	30.8 ± 1.7 *^	34.8 ± 1.9 *	32.1 ± 2.3 ^	26.1 ± 1.8
VK + Extract B 40 mg/kg	29.2 ± 1.7	28.2 ± 0.8	29.9 ± 1.7	26.5 ± 0.7	26.2 ± 2.2

* statistically significant differences compared to VK (*p* < 0.05); ^—statistically significant differences for the respective group compared to t = 0 (*p* < 0.05); VK—Vincristine—0.1 mg/kg of body weight, intraperitoneal administration; Extract D—*Cannabis sativa* L. extract, *variety Dora*; the dose expressed as synthetic Cannabidiol, oral administration; Extract B—*Cannabis sativa* L. extract, *variety Tygra;* the dose expressed as synthetic Cannabidiol, oral administration; i.p.—intraperitoneal; p.o.—per os.

**Table 7 biomolecules-14-01065-t007:** The CB1R and CB2R expression in the brain cortex, hippocampus, and lymphocytes in response to treatment with vincristine vs. NaCl and gabapentin vs. vincristine.

CB1R (Vincristine Group VK vs. NaCl Group)					
Intervention	Hippocampus		Cortex		Lymphocytes
	Protein [ng/mL] Median (1st; 3rd Quartile)	Gene Expression Median (1st; 3rd Quartile)	Protein [ng/mL] Median (1st; 3rd Quartile)	Gene Expression Median (1st; 3rd Quartile)	Gene Expression Median (1st; 3rd Quartile)
NaCl- 1 mL i.p. + Rape oil -1 mL p.o.	1.45 (1.14; 2.69)	3.04 × 10^−3^ (1.92 × 10^−3^; 5.25 × 10^−3^)	3.99 (3.32; 4.57)	21.20 × 10^−3^ (13.10 × 10^−3^; 84.20 × 10^−3^)	0.03 × 10^−3^ (0.001 × 10^−3^; 0.87 × 10^−3^)
VK +Rape oil—1 mL p.o.	0.89 (0.39; 1.03)	4.36 × 10^−3^ (2.92 × 10^−3^; 6.68 × 10^−3^)	2.75 (2.36; 3.78) *	22.90 × 10^−3^ (14.40 × 10^−3^; 35.40 × 10^−3^)	0.02 × 10^−3^ (0.05 × 10^−3^; 0.16 × 10^−3^)
**CB2R (vincristine group VK vs. NaCl group)**					
NaCl- 1 mL i.p. + Rape oil -1 mL p.o.	0.25 (0.18; 0.32)	3.90 × 10^−3^ (2.21 × 10^−3^; 11.40 × 10^−3^)	1.26 (1.02; 1.45)	50.30 × 10^−3 ^ (13.80 × 10^−3^; 71.40 × 10^−3^)	32.10 × 10^−3^ (14.60 × 10^−3^; 41.50 × 10^−3^)
VK + Rape oil—1 mL p.o.	0.29 (0.26; 0.55)	3.45 × 10^−3^ (2.71 × 10^−3^; 4.56 × 10^−3^)	0.87 (0.83; 0.97)	3.18 × 10^−3^ * (1.67 × 10^−3^; 7.43 × 10^−3^)	23.50 × 10^−3 ^ (18.70 × 10^−3^; 84.30 × 10^−3^)
**CB1R (vincristine group VK vs. gabapentin group)**					
VK + Rape oil—1 mL p.o.	0.89 (0.39; 1.03)	4.36 × 10^−3^ (2.92 × 10^−3^; 6.68 × 10^−3^)	2.75 (2.36; 3.78)	22.90 × 10^−3^ (14.40 × 10^−3^; 35.40 × 10^−3^)	0.02 × 10^−3^ (0.05 × 10^−3^; 0.16 × 10^−3^)
VK + Gabapentin-60 mg/kg p.o.	2.51 (2.13; 2.8) ^	3.02 × 10^−3^ (1.14 × 10^−3^; 4.6 × 10^−3^)	2.74 (2.09; 3.2)	31.30 × 10^−3 ^ (26.50 × 10^−3^; 33.30 × 10^−3^)	0.01 × 10^−3 ^ (0.001 × 10^−3^; 0.011 × 10^−3^)
**CB2R (vincristine group VK vs. gabapentin group)**					
VK +Rape oil—1 mL p.o.	0.29 (0.26; 0.55)	3.45 × 10^−3^ (2.71 × 10^−3^; 4.56 × 10^−3^)	0.87 (0.83; 0.97)	3.18 × 10^−3 ^ (1.67 × 10^−3^; 7.43 × 10^−3^)	23.50 × 10^−3^ (18.70 × 10^−3^; 84.30 × 10^−3^)
VK + Gabapentin-60 mg/kg p.o.	0.42 (0.34; 0.5)	4.79 × 10^−3^ (2.94 × 10^−3^; 13.80 × 10^−3^)	1.25 (1.03; 1.40)	2.35 × 10^−3^ (0.51 × 10^−3^; 3.63 × 10^−3^)	35.30 × 10^−3 ^ (14.70 × 10^−3^; 57.10 × 10^−3^)

* statistically significant differences compared to NaCl (*p* < 0.05); ^—statistically significant compared to vincristine VK (*p* < 0.05); VK—Vincristine—0.1 mg/kg of body weight, intraperitoneal administration Extract D—*Cannabis sativa* L. extract, *variety Dora*; the dose expressed as synthetic Cannabidiol, oral administration Extract B—*Cannabis sativa* L. extract, *variety Tygra;* the dose expressed as synthetic Cannabidiol, oral administration i.p.—intraperitoneal; p.o.—per os.

**Table 8 biomolecules-14-01065-t008:** CB1R expression in the brain cortex, hippocampus, and lymphocytes in response to vincristine, gabapentin, and CSL extracts.

CB1R					
Type of Intervention	Hippocampus		Cortex		Lymphocytes
	Protein [ng/mL] Median (1st; 3rd Quartile)	Gene Expression Median (1st; 3rd Quartile)	Protein [ng/mL] Median (1st; 3rd Quartile)	Gene Expression Median (1st; 3rd Quartile)	Gene Expression Median (1st; 3rd Quartile)
VK + Rape oil	0.89 (0.39; 1.03)	4.36 × 10^−3^ (2.92 × 10^−3^; 6.68 × 10^−3^)	2.75 (2.36; 3.78)	22.90 × 10^−3^ (14.40 × 10^−3^; 35.40 × 10^−3^)	0.02 × 10^−3 ^ (0.05 × 10^−3^; 0.16 × 10^−3^)
VK + Gabapentin	2.51 (2.13; 2.8)	3.02 × 10^−3^ (1.14 × 10^−3^; 4.6 × 10^−3^)	2.74 (2.09; 3.2)	31.30 × 10^−3 ^ (26.50 × 10^−3^; 33.30 × 10^−3^)	0.01 × 10^−3 ^ (0.001 × 10^−3^; 0.011 × 10^−3^)
VK + Extract D 5 mg/kg	1.96 (1.44; 2.09)	3.15 × 10^−3^ (1.45 × 10^−3^; 3.67 × 10^−3^)	2.97 (1.74; 3.2)	2.95 × 10^−3 ^ (2.21 × 10^3^; 6.61 × 10^−3^)	^15^ 0.07 × 10^−3 ^ (0.01 × 10^−3^; 0.10 × 10^−3^)
VK + Extract D 7.5 mg/kg	2.4 (2.17; 3.8)	^5^ 1.73 × 10^−3^ (0.89 × 10^−3^; 2.43 × 10^−3^)	^6^ 3.4 (3.27; 4.2)	^11^ 439.90 × 10^−3^ (2.72 × 10^−3^; 1957 × 10^−3^)	0.11 × 10^−3 ^ (0.06 × 10^−3^; 0.18 × 10^−3^)
VK + Extract D 10 mg/kg	^1^ 3.26 (3.22; 4.06)	9.97 × 10^−3^ (5.51 × 10^−3^; 14.7 × 10^−3^)	^7^ 4.35 (4.11; 5.15)	5.51 × 10^−3 ^ (3.56 × 10^−3^; 5.51 × 10^−3^)	0.64 × 10^−3 ^ (0.16 × 10^−3^; 1.08 × 10^−3^)
VK + Extract D 20 mg/kg	2.24 (1.71; 2.95)	6.54 × 10^−3^ (0.37 × 10^−3^; 7.35 × 10^−3^)	1.72 (1.06; 2.24)	^12^ 0.24 × 10^−3^ (0.03 × 10^−3^; 1.53 × 10^−3^)	0.56 × 10^−3 ^ (0.03 × 10^−3^; 2.03 × 10^−3^)
VK + Extract D 40 mg	^2^ 3.77 (2.78; 4.34)	2.44 × 10^−3^ (0.89 × 10^−3^; 3.8 × 10^−3^)	1.5 (1.35; 1.77)	4.12 × 10^−3^ (2.34 × 10^−3^; 5.64 × 10^−3^)	0.14 × 10^−3 ^ (0.01 × 10^−3^; 0.21 × 10^−3^)
VK + Extract B 5 mg/kg	1.24 (0.26; 2.65)	1.57 × 10^−3^ (0.70 × 10^−3^; 5.85 × 10^−3^)	1.53 (1.16; 1.64)	^13^ 1.14 × 10^−3^ (0.34 × 10^−3^; 2.22 × 10^−3^)	0.26 × 10^−3 ^ (0.12 × 10^−3^; 0.74 × 10^−3^)
VK + Extract B 7.5 mg/kg	^3^ 5.65 (3.22; 9.02)	7.57 × 10^−3^ (5.19 × 10^−3^; 11.7 × 10^−3^)	1.49 (1.19; 1.55)	^14^ 0.05 × 10^−3^ (0.001 × 10^−3^; 0.07 × 10^−3^)	0.41 × 10^−3 ^ (0.11 × 10^−3^; 1.29 × 10^−3^)
VK + Extract B 10 mg/kg	^4^ 3.6 (2.90; 5.00)	8.12 × 10^−3^ (4.17 × 10^−3^; 12.0 × 10^−3^)	^8^ 0.6 (0.2; 1.0)	11.16 × 10^−3^ (9.82 × 10^−3^; 19.30 × 10^−3^)	0.27 × 10^−3 ^ (0.11 × 10^−3^; 0.54 × 10^−3^)
VK + Extract B 20 mg/kg	2.88 (2.70; 3.60)	4.17 × 10^−3^ (3.41 × 10^−3^; 5.07 × 10^−3^)	^9^ 0.36 (0.34; 0.4)	2.43 × 10^−3^ (1.73 × 10^−3^; 3.55 × 10^−3^)	0.37 × 10^−3 ^ (0.11 × 10^−3^; 0.61 × 10^−3^)
VK + Extract B 40 mg/kg	2.44 (1.43; 3.38)	7.99 × 10^−3^ (1.90 × 10^−3^; 9.59 × 10^−3^)	^10^ 0.75 (0.53; 1.01)	3.77 × 10^−3^ (1.86 × 10^−3^; 5.46 × 10^−3^)	0.02 × 10^−3^ (0.01 × 10^−3^; 0.04 × 10^−3^)

^1^ VK + Extract D 10 mg/kg vs. VK (*p* < 0,05); VK + Extract D 10 mg/kg vs. VK + Extract D 5 mg/kg (*p* < 0,05); ^2^ VK + Extract D 40 mg vs. VK (*p* < 0,05); VK + Extract D 40 mg vs. VK + Extract D 5 mg/kg (*p* < 0,05); ^3^ VK + Extract B 7.5 mg/kg vs. VK (*p* < 0,001); ^4^ VK + Extract B 10 mg/kg vs. VK (*p* < 0,05); ^5^ VK + Extract D 7.5 mg/kg vs. VK + Extract D 10 mg/kg (*p* < 0.05); ^6^ VK + Extract D 7.5 mg/kg vs. VK + Extract D 40 mg/kg (*p* < 0.05); ^7^ VK + Extract D 10 mg/kg vs. VK + Extract D 20 mg/kg (*p* < 0.05); VK + Extract D 10 mg/kg vs. VK + Extract D 40 mg/kg (*p* < 0.05); ^8^ VK + Extract B 10 mg/kg vs. VK (*p* < 0,05); ^9^ VK + Extract B 20 mg/kg vs. VK (*p* < 0,001); ^10^ VK + Extract B 40 mg/kg vs. VK (*p* < 0,05); ^11^ VK + Extract D 7.5 mg/kg vs. VK + Extract D 20 mg/kg (*p* < 0.05); ^12^ VK + Extract D 20 mg/kg vs. VK (*p* < 0,05); ^13^ VK + Extract B 5 mg/kg vs. VK (*p* < 0,05); ^14^ VK + Extract B 7.5 mg/kg vs. VK (*p* < 0,001); ^15^ VK + Extract D 5 mg/kg vs. VK + Extract D 10 mg/kg (*p* < 0,05); VK—Vincristine—0.1 mg/kg of body weight, intraperitoneal administration Extract D—*Cannabis sativa* L. extract, *variety Dora*; the dose expressed as synthetic Cannabidiol, oral administration Extract B—*Cannabis sativa* L. extract, *variety Tygra*; the dose expressed as synthetic Cannabidiol, oral administration i.p.–intraperitoneal; p.o.–per os.

**Table 9 biomolecules-14-01065-t009:** CB2 receptor expression in the brain cortex, hippocampus, and lymphocytes in response to vincristine, gabapentin, and CSL extracts.

Type of Intervention	CB2R				
	Hippocampus		Cortex		Lymphocytes
	Protein [ng/mL] Median (1st; 3rd Quartile)	Gene Expression Median (1st; 3rd Quartile)	Protein [ng/mL] Median (1st; 3rd Quartile)	Gene Expression Median (1st; 3rd Quartile)	Gene Expression Median (1st; 3rd Quartile)
VK + Rape oil	0.29 (0.26; 0.55)	3.45 × 10^−3^ (2.71 × 10^−3^; 4.56 × 10^−3^)	0.87 (0.83; 0.97)	3.18 × 10^−3^ (1.67 × 10^−3^; 7.43 × 10^−3^)	23.50 × 10^−3^ (18.70 × 10^−3^; 84.30 × 10^−3^)
VK + Gabapentin	0.42 (0.34; 0.5)	4.79 × 10^−3^ (2.94 × 10^−3^; 13.80 × 10^−3^)	1.25 (1.03; 1.40)	2.35 × 10^−3^ (0.51 × 10^−3^; 3.63 × 10^−3^)	35.30 × 10^−3^ (14.70 × 10^−3^; 57.10 × 10^−3^)
VK + Extract D 5 mg/kg	0.25 (0.22; 0.41)	2.79 × 10^−3^ (2.02 × 10^−3^; 5.33 × 10^−3^)	0.97 (0.87; 1.13)	13.90 × 10^−3^ (6.15 × 10^−3^; 29.60 × 10^−3^)	43.80 × 10^−3^ (32.70 × 10^−3^; 117.30 × 10^−3^)
VK + Extract D 7.5 mg/kg	0.4 (0.27; 0.50)	^2^ 7.87 × 10^−3^ (4.51 × 10^−3^; 22.9 × 10^−3^)	1.00 (0.92; 1.10)	27.90 × 10^−3^ (15.00 × 10^−3^; 49.70 × 10^−3^)	59.00 × 10^−3 ^ (29.10 × 10^−3^; 79.60 × 10^−3^)
VK + Extract D 10 mg/kg	0.32 (0.29 0.41)	1.49 × 10^−3^ (1.32 × 10^−3^; 3.72 × 10^−3^)	^3^ 1.06 (0.88; 1.42)	^6^ 3.24 × 10^−3^ (2.34 × 10^−3^; 7.20 × 10^−3^)	61.50 × 10^−3^ (52.40 × 10^−3^; 91.90 × 10^−3^)
VK + Extract D 20 mg/kg	0.25 (0.12; 0.29)	2.39 × 10^−3^ (0.35 × 10^−3^; 7.03 × 10^−3^)	0.76 (0.71; 0.88)	45.50 × 10^−3^ (13.50 × 10^−3^; 97.20 × 10^−3^)	60.10 × 10^−3^ (39.80 × 10^−3^; 73.10 × 10^−3^)
VK + Extract D 40 mg	0.39 (0.34; 0.44)	5.73 × 10^−3^ (1.36 × 10^−3^; 8.47 × 10^−3^)	0.34 (0.25; 0.72)	5.50 × 10^−3^ (4.00 × 10^−3^; 15.50 × 10^−3^)	38.20 × 10^−3^ (32.00 × 10^−3^; 84.60 × 10^−3^)
VK + Extract B 5 mg/kg	0.32 (0.22; 0.45)	2.31 × 10^−3^ (1.06 × 10^−3^; 3.02 × 10^−3^)	0.27 (0.13; 0.41)	14.50 × 10^−3^ (6.01 × 10^−3^; 27.70 × 10^−3^)	29.20 × 10^−3^ (15.40 × 10^−3^; 73.80 × 10^−3^)
VK + Extract B 7.5 mg/kg	^1^ 1.39 (0.83; 1.52)	0.27 × 10^−3^ (0.20 × 10^−3^; 0.42 × 10^−3^)	0.10 (0.08; 0.17)	0.32 × 10^−3^ (0.001 × 10^−3^; 0.13 × 10^−3^)	37.40 × 10^−3^ (25.30 × 10^−3^; 47.40 × 10^−3^)
VK + Extract B 10 mg/kg	0.8 (0.70; 1.00)	0.30 × 10^−3^ (0.16 × 10^−3^; 0.48 × 10^−3^)	^4^ 0.10 (0.01; 0.19)	1.62 × 10^−3^ (0.99 × 10^−3^; 1.87 × 10^−3^)	63.80 × 10^−3^ (31.70 × 10^−3^; 72.50 × 10^−3^)
VK + Extract B 20 mg/kg	0.94 (0.73; 1.10)	1.51 × 10^−3^ (0.53 × 10^−3^; 3.40 × 10^−3^)	^5^ 0.05 (0.04; 0.10)	1.53 × 10^−3^ (1.21 × 10^−3^; 2.89 × 10^−3^)	24.00 × 10^−3 ^ (19.30 × 10^−3^; 38.60 × 10^−3^)
VK + Extract B 40 mg/kg	0.92 (0.54; 1.02)	5.82 × 10^−3^ (1.36 × 10^−3^; 6.96 × 10^−3^)	0.09 (0.07; 0.12)	1.00 × 10^−3^ (0.65 × 10^−3^; 1.34 × 10^−3^)	48.20 × 10^−3 ^ (42.00 × 10^−3^; 54.90 × 10^−3^)

^1^ VK + Extract B 7.5 mg/kg vs. VK (*p* < 0.05); ^2^ VK + Extract D 7.5 mg/kg vs. VK + Extract D 10 mg/kg (*p* < 0.05); ^3^ VK + Extract D 10 mg/kg vs. VK + Extract D 40 mg/kg vs. (*p* < 0.05); ^4^ VK + Extract B 10 mg/kg vs. VK (*p* < 0.001); ^5^ VK + Extract B 20 mg/kg vs. VK (*p* < 0.05); ^6^ VK + Extract D 10 mg/kg vs. VK + Extract D 20 mg/kg vs. (*p* < 0.05); RT-PCR-Real Time PCR ELISA—enzyme-linked immunosorbent assay Vin- Vincristine—0.1 mg/kg of body weight, intraperitoneal administration Extract D—*Cannabis sativa* L. extract, *variety Dora*; the dose expressed as synthetic Cannabidiol, oral administration Extract B—*Cannabis sativa* L. extract, *variety Tygra*; the dose expressed as synthetic Cannabidiol, oral administration i.p.– intraperitoneal; p.o.—per os; bw—body weight.

## Data Availability

The data presented in this study are available on request from the corresponding author due to an ongoing patent process related to the development of a medicinal product used to support the treatment of neuropathic pain, in accordance with the agreements of the research project INNOMED/I/11/NCBR/2014.

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
