# Peer review of "Cannabis sativa L. Extract Alleviates Neuropathic Pain and Modulates CB1 and CB2 Receptor Expression in Rat"

_biomolecules, 2024, doi:10.3390/biom14091065_

Round 1

Reviewer 1 Report

Comments and Suggestions for Authors

1- The study uses male Wistar rats and induces neuropathic pain with vincristine, followed by treatment with gabapentin or Cannabis sativa extracts. However, the manuscript does not provide sufficient justification for the chosen dosages of vincristine and gabapentin, nor does it explain why only male rats were used.

2- The manuscript presents results from behavioral tests and receptor expression analyses, it lacks detailed information on the statistical methods used. For example, whether appropriate post-hoc tests were conducted following ANOVA to determine the significance between specific groups is unclear.

3- The conclusion suggests that alterations in cannabinoid receptor expression do not fully explain the observed behavioral changes and that additional signaling pathways may be involved. However, the manuscript does not explore these potential pathways in depth.

Author Response

Comments 1: The study uses male Wistar rats and induces neuropathic pain with vincristine, followed by treatment with gabapentin or Cannabis sativa extracts. However, the manuscript does not provide sufficient justification for the chosen dosages of vincristine and gabapentin, nor does it explain why only male rats were used.

Response 1: Thank you for your suggestion. Regarding the dosages of vincristine and gabapentin used in our study, we based these on established literature and models relevant to our research. Based on in vivo studies conducted using a rat model, a dose of 0.1 mg/kg vincristine has been shown to effectively induce peripheral neuropathy [Chiba; Ja'afer; Jaggi; Han, Aley]. Gabapentin is a recommended treatment for neuropathic pain, with its efficacy having been repeatedly confirmed in in vivo studies using male Wistar rats. In our study, gabapentin was administered at a dose of 60 mg/kg body weight, which is consistent with the literature that identifies this dose as effective in models of chemotherapy-induced neuropathic pain [Folkesson; Mangaiarkkarasi]. As for the use of male Wistar rats exclusively, this decision was informed by the prevailing literature, which often employs male rodents in neuropathic pain studies due to their more consistent response patterns and the need to minimize variability. Our study's primary aim was to evaluate the pharmacological activity of the treatments, rather than to assess potential sex differences in pain response. Consequently, we followed common practice by using male rats to ensure uniformity and reliability in our results.The revised sections have been highlighted in yellow in the Discussion.

Comments 2: The manuscript presents results from behavioral tests and receptor expression analyses, it lacks detailed information on the statistical methods used. For example, whether appropriate post-hoc tests were conducted following ANOVA to determine the significance between specific groups is unclear.

Response 2: Thank you for your valuable feedback. We have added detailed information regarding the statistical analysis to the manuscript, including the post-hoc tests conducted following ANOVA to determine significance between specific groups. These updates have been highlighted in yellow for your convenience.

Comments 3: The conclusion suggests that alterations in cannabinoid receptor expression do not fully explain the observed behavioral changes and that additional signaling pathways may be involved. However, the manuscript does not explore these potential pathways in depth.

Response 3: Thank you for your valuable comment regarding the need to address the role of other potential molecular pathways in the modulation of pain influenced by cannabis. We have revised our conclusions to include other mechanisms, such as TRPV1, which may also play a role in neuropathic pain. Our study primarily focused on the expression of cannabinoid receptors CB1R and CB2R in the context of neuropathy to investigate their involvement at the molecular level. We would like to emphasize that our analysis aimed to explore the expression of cannabinoid receptors CB1R and CB2R in neuropathic pain and assess their molecular involvement. We have highlighted that changes in cannabinoid receptors may not fully account for the behavioral changes observed in response to neuropathic pain, and additional signaling pathways might contribute to the initiation and transmission of neuropathic pain. The revised sections have been highlighted in yellow.

Reviewer 2 Report

Comments and Suggestions for Authors

This work presents an experimental design partially suitable for carrying out what was proposed in its objective. It uses an adequate positive control to verify the anti-neuropathic treatment. The number of animals used to conduct the experiments makes the results obtained interesting as well as the verification of the expression of receptors in the central nervous system. However, some modifications are necessary to be made before publishing:

1.       Please checked all the abbreviations trough the manuscript

2.       Throughout the text, the authors talk about analgesic activity, especially in relation to animals, we evaluated antinociceptive activity against or nociception generated by vincristine, so these terms must be replaced throughout the article.

3.       In the introduction there is a long part that talks about the activity of cannabinoids in general, which seems to be somewhat lost in the context of the work. So it seems necessary for this part to be rewritten and addressed, but the antinociceptive part and the mechanisms that are already understood about Cannabis sativa in neuropathic-like pai models needs to be more explored.

4.       Despite the authors demonstrating their experimental protocol in the figure 1, it is not clear, whether prior to the administration of vincristine, if the baseline of the nociceptive behavior from the animals were measure.

5.       There is some report that cannabinoid receptors can be activated by thermal hyperalgesia? If this report does not exist, it is necessary to discuss this topic and describe which other receptors may be affected by the article's thermal hyperalgesia.

6.       Another important factor to be discussed is why some higher doses have no effect and lower doses have an effect and this should be expected. Another factor that could be discussed or addressed is why some doses generated an increase in receptor expression and others generated admission and what the possible implications of these results.

Comments on the Quality of English Language

English writing can be improved

Author Response

Comments 1: Please checked all the abbreviations trough the manuscript

Response1: Thank you for your valuable insights and for bringing this to our attention. We have carefully reviewed and corrected the abbreviations throughout the manuscript, with particular focus on the cannabinoid receptors. The nomenclature has now been consistently standardized across the entire text.

Comments 2: Throughout the text, the authors talk about analgesic activity, especially in relation to animals, we evaluated antinociceptive activity against or nociception generated by vincristine, so these terms must be replaced throughout the article.

Response 2: Thank you for your valuable suggestion. In accordance with your recommendation, we have revised the manuscript to replace the term "analgesic activity" with "antinociceptive activity" throughout the text, particularly in relation to the nociception generated by vincristine.

Comments 3: In the introduction there is a long part that talks about the activity of cannabinoids in general, which seems to be somewhat lost in the context of the work. So it seems necessary for this part to be rewritten and addressed, but the antinociceptive part and the mechanisms that are already understood about Cannabis sativa in neuropathic-like pai models needs to be more explored.

Response 3: Thank you for your valuable feedback. We have revised the Introduction to focus specifically on the antinociceptive activity of cannabinoids and the mechanisms that are well understood in the context of Cannabis sativa in neuropathic pain models. We have rewritten the relevant sections to better align with the context of our study and highlighted the revised and expanded passages in yellow. We hope that these changes will meet with the reviewer's approval.

Comments 4: Despite the authors demonstrating their experimental protocol in the figure 1, it is not clear, whether prior to the administration of vincristine, if the baseline of the nociceptive behavior from the animals were measure.

Response 4: Thank you for your valuable feedback. We have updated the Materials and Methods as well as the Discussion sections of the manuscript to clarify the use of the control group. In experimental models involving healthy animals before the induction of neuropathic pain, it is assumed a priori that these animals do not experience the nociception characteristic of pain induced by pharmacological substances and do not exhibit a pathological pain state. Therefore, we employed a negative control group, where animals received NaCl instead of substances inducing a pathological condition. These animals were also administered the test substances, allowing for a precise evaluation of their effects without induced pain. The revised sections have been highlighted in yellow.

Comments 5: There is some report that cannabinoid receptors can be activated by thermal hyperalgesia? If this report does not exist, it is necessary to discuss this topic and describe which other receptors may be affected by the article's thermal hyperalgesia.

Response 5: Thank you for your valuable comment. We appreciate your attention to the issue regarding cannabinoid receptors and thermal hyperalgesia.There is currently no conclusive evidence suggesting that cannabinoid receptors are activated by thermal hyperalgesia. However, the perception of pain and temperature is closely linked through the action of specific receptors that respond to thermal stimuli. For example, TRPV1 receptors are known to respond to high temperatures perceived as painful and play a crucial role in converting thermal information into pain sensations [Andolfo]. We have added this discussion to the Discussion section of the manuscript and highlighted it in yellow for ease of reference. Furthermore, we would like to note that the hot plate test used in our experiment is a widely accepted method for assessing pain responses in animals, similar to the tail flick test. Both of these tests are commonly employed to evaluate the efficacy of analgesic drugs.

Comments 6: Another important factor to be discussed is why some higher doses have no effect and lower doses have an effect and this should be expected. Another factor that could be discussed or addressed is why some doses generated an increase in receptor expression and others generated admission and what the possible implications of these results.

Response 6: Thank you for your insightful comments.Of course, our molecular expression analysis at both the mRNA and protein levels did not confirm a dose-dependent effect of Cannabis extracts. While the expectation of dose-dependency is valid for single active substances, plant extracts, which contain multiple active compounds, often exhibit variable and biphasic effects that are not always dose-dependent [Russo; Russo and Taming; Shustorovich et al., 2024]. This variability is also reflected in cannabinoid receptor expression levels, which we observed as independent of the dose used.

Furthermore, the primary function of the endocannabinoid system is to maintain homeostasis by modulating cannabinoid receptor activity to regulate physiological processes [Lowe]. The pleiotropic nature of Cannabis extracts can lead to complex interactions at the molecular level. For example, CBD alone has been shown to affect the expression of approximately 680 genes, whereas THC affects only 58 genes. This complexity is compounded by the interaction of cannabinoids with various signaling pathways and factors, highlighting the need for in-depth molecular investigations to understand the dose-dependent interactions within the intricate network of cannabinoid effects [Juknat et al., 2012; Blando et al., 2022].

We have incorporated these detailed explanations into the Discussion section of our manuscript and highlighted them in yellow.

We would like to inform you that a professional language editing service has reviewed the manuscript, and we have obtained a certificate confirming that a native English speaker has carefully proofread the text to enhance its clarity and coherence.

Round 2

Reviewer 2 Report

Comments and Suggestions for Authors

I appreciate the clarification of my doubts. After review, the manuscript can be accepted, although the authors did not show the data on the basal threshold of nociception. I understand that the experiments have already been conducted and that they would need to be repeated to verify the basal threshold. The use of the untreated control can be justified, although it does not represent the basal threshold of the animals. Therefore, my last suggestion is to describe in the methodology that the basal threshold of nociception of the animals was not performed. Therefore, in this study, the presence and influence of previous nociception in the groups that were subjected to the administration of vincristine was not verified.

Author Response

Dear Reviewer,

Thank you very much for your valuable feedback. We have updated the methodology section of the manuscript to include information regarding the absence of a basal nociception threshold measurement in the animals. We acknowledge the importance of this aspect and will make it a priority in future experiments. Should we obtain approval from the Ethics Committee, we will incorporate such measurements in subsequent studies to better assess the presence and impact of pre-existing nociception in the vincristine-treated groups.

We appreciate your understanding and thoughtful suggestion, and we hope that the revised manuscript meets your expectations.

Sincerely,

Joanna Bartkowiak-Wieczorek
